# RSS/TDoA-Based Source Localization in Microwave UWB Sensors Networks Using Two Anchor Nodes [note 1]

**DOI:** 10.3390/s22083018

**Published:** 2022-04-14

**Authors:** Sergei Ivanov, Vladimir Kuptsov, Vladimir Badenko, Alexander Fedotov

**Affiliations:** Institute of Electronics and Telecommunications, Peter the Great St. Petersburg Polytechnic University, 29 Polytechnicheskaya Str., 195251 Saint Petersburg, Russia; kuptsov@spbstu.ru (V.K.); badenko_vl@spbstu.ru (V.B.); afedotov@spbstu.ru (A.F.)

**Keywords:** localization, UWB Networks, maximum likelihood (ML) estimator, received signal strength (RSS), time difference of arrival (TDOA)

## Abstract

The manuscript presents an algorithm for the optimal estimation of the amplitude and propagation delay time of an ultra-wideband radio signal, in systems for the passive location of fixed targets based on the hybrid RSS/TDoA method in two-dimensional space with two base stations. The optimal estimate is based on the Bayesian strategy of maximum a posteriori probability density, taking into account a priori data on the statistical properties of the Line of Sight radio channel during Gaussian monocycle propagation. The Bayesian Cramer–Rao lower bound (BCRLB) of the delay time and the amplitude estimates for a time-discrete signal are calculated, and the resulting parameter estimate is compared with BCRLB. An algorithm has been developed for optimal estimation of distances from the radiation source to base stations, based on the results of the measurements of the amplitude and the propagation delay time of the UWB radio signal. The calculation of the statistical characteristics of the obtained estimate is carried out, and the functional dependence of the characteristics on various parameters is analyzed.

## 1. Introduction

The coordinates determination of a radio emission source using Microwave Sensors Networks (MSNs) is widely used in devices for various applications, including environmental monitoring, intelligent transportation, precision agriculture, smart home, medicine, and others [1,2,3,4,5]. For some sensor nodes, called anchor nodes (AN) (base stations), spatial coordinates are known in some positioning system. The location coordinates of the remaining nodes, called source nodes (SN), are estimated using various positioning algorithms, usually based on measuring radio signal parameters [1,6].

At present, several efficient methods for the spatial positioning of radio sources using MSNs have been developed. These include the following positioning techniques:time of arrival (TOA) [7,8,9],time-difference-of-arrival (TDOA) [2,10,11],direction of arrival (DOA) or angle of arrival (AOA) [1],received signal strength (RSS) indicator [12,13,14].

Each of the listed positioning techniques has its advantages and disadvantages and is used to solve specific problems in given conditions. It is effective to combine several positioning techniques, for example, TDOA and FDOA [15] or TDOA and AOA [16]. In this work, we analyze a hybrid method for positioning a quasi-stationary radio source using TDOA/RSS algorithms for processing measurements.

When the location system is operating in RSS mode, the power *P* or amplitude *A* of the signal emitted by the source is measured at the input of the base station receiver. In a real radio channel, the signal level is attenuated due to three factors: large-scale path loss, multipath fading, and shadowing [17]. In free space, large-scale path loss is a deterministic function of the distance between transmitter (Tr) and receiver (Rv), proportional to *r*^−2^, where *r* is the distance between Tr and Rv. Multipath fading is caused by interference at the receiving point of several copies of the transmitted signal Tr that are delayed for different times. One effective way to deal with multipath fading is to use ultra wideband (UWB) pulsed radio systems, due to their high time resolution capability [18]. However, UWB technology is not yet fully developed, and efficient design of such communication systems requires new experimental and theoretical studies. In particular, it is necessary to develop a statistical channel model that adequately describes the propagation of a UWB signal under various conditions, but, at the same time, it should be simple enough to make theoretical analysis and computer simulation possible. The use of such an analytical model (in a closed form), for constructing a localization algorithm and calculating the statistical characteristics of estimating the position of a radio source using MSNs, is not currently described in the literature.

As confirmed by numerous studies, the most-tested statistical model that reflects the small-scale properties of the UWB channel, including in indoor spaces with small volume, is the modified Saleh–Valenzuela parametric model [18], which, as it turned out, corresponded best of all to the experimental data [19,20,21,22].

Statistical small-scale properties of a discrete internal radio channel are reflected by the impulse response *h*(*t*), which for the modified Saleh–Valenzuela model has the form [23,24].
(1)$$h\left(t\right)=X\overset{L}{\underset{l=1}{\sum }}\overset{K}{\underset{k=1}{\sum }}\beta_{k,l}\delta \left(t-T_{l}-\tau_{k,l}\right),\;\;$$
where *T*_l_ is the signal propagation time delay of the *l*-th cluster, β*_k,l_* and τ*_k,l_* are the amplitude and arrival delay of the *k*-th ray in the *l*-th cluster, *L* is the number of clusters, and *K* is the maximum number of rays in the cluster. Time delays *T*_l_ and τ*_k,l_* are described by a Poisson process, while the random variable *X* is called the shading factor and is described by a lognormal distribution with a standard deviation σ_X_, called the shading depth.

Numerous studies have shown that for the amplitude β*_k,l_*, the lognormal or Nakagami distributions provide the best of all correspondence to the experimental data [23,24,25]. As follows from (1), the statistical characteristics of the UWB channel of MSNs differ significantly from the characteristics of a narrowband channel. There are also some peculiarities of the propagation of UWB signals at indoor spaces with small volume. We will use the statistical properties of *h*(*t*) to develop an algorithm for efficient estimation of the spatial position of SN and to calculate the Cramer–Rao lower bound for this estimation.

In [7,26,27], the results of the positioning coordinates estimation obtained by the TDoA and RSS methods are compared. It is shown that the efficiency of TDoA-based systems increases when using UWB signals and signals with a large base (the product of duration and signal bandwidth). However, TDoA positioning errors increase dramatically when MSNs are operated indoors, where source signal undetectable direct path (UDP) conditions occur due to the presence of a large number of NLOS channels or a large attenuation of the signal when it propagates along the LOS channel. The efficiency of RSS-based systems is less sensitive to operating conditions in large-scale path loss, multipath fading, and shadowing modes, but signal strength estimation methods have large errors.

In works [28,29,30,31], in contrast to existing studies, in which it was assumed that the TDOA measurement noise variance was independent of the respective distances from the source to the sensor, a more realistic model was considered, in which the signal-to-noise ratio (SNR) is a function of the distance from the source to the sensor. The proposed hybrid TDoA/RSS methods improve the accuracy of source localization estimation compared to TDOA or RSS methods used separately. The gain of the hybrid TDoA/RSS method is especially noticeable in the proximity to the base stations but makes a small contribution at a low noise level. It is shown that the estimation accuracy of the proposed localization algorithm is comparable to the Cramer–Rao lower bound (CRLB), taking into account moderate TDOA measurement noise. However, the results obtained in these works do not consider the specifics of UWB signals. The statistical properties of the signal propagation channel in MSNs are taken into account by the log-normal distribution of the integral (total) power of the received signal at the SN input. This does not take into account the multipath components of NLOS and LOS, and the First Detected Peak of the Direct Path channel is not determined.

A method for determining the coordinates of a radiation source in an indoor space, taking into account the decomposition of the signal in the statistical channel of MSNs into multipath components, was proposed in [32,33,34]. As part of the received signal at the SN input, the NLOS and LOS components are distinguished. Using the statistical properties of the signal propagation channel, the FDP is identified with the DP channel or the Undetected Direct Path (UDP) mode is registered. In the first case, the SN coordinates are determined by the FDP time position using the ToA or TDoA method; in the second case, we can talk about a possible large localization error. The proposed semi-empirical target localization technique makes it difficult to carry out a theoretical analysis of the statistical properties of the SN coordinate estimate, and, therefore, such an analysis is not carried out in the above works.

A large number of works are devoted to the development of methods for determining the type of the probability distribution function and estimating its parameters that describe the statistical properties of the propagation of signals (including ultra-wideband signals) in MSNs channels in indoor spaces (see, for example, [34,35,36]). The resulting estimates and properties of parameters and distributions are used to develop methods for efficient estimation of SN location coordinates.

The development of hybrid TDoA/RSS positioning methods using UWB signals for MSNs in indoor spaces has been the subject of many studies, and the results of some are presented in [37,38,39,40,41,42,43,44,45]. An analysis of these results shows that in the absence of a priori experimental data on the statistical characteristics of signal propagation channels in this particular heterogeneous environment, localization based on joint RSS/TDoA methods (strategies) is a complex problem that is difficult to solve without simplifying the mathematical model. For example, to obtain a strategy estimate for the maximum likelihood function or weighted least squares RWLS related to a non-convex optimization problem, a transformation to a convex optimization problem is used, which is then solved by methods of semi-definite programming SDP.

The objective of the study is based on the following statements. As the analysis of the results of the work shows, the problem of estimating the coordinates of the radiation source is solved by complex numerical calculations. The solution is multiparametric, but not all dependencies can be obtained and have a physical interpretation. At the same time, there is a practical need to use a small (minimal) number of ANs in MSNs. It is also possible to operate MSNs in conditions where the influence of NLOS channels on LOS channels is small, for example, when using UWB signals and the appropriate operating range of base station receivers. However, at the same time, the power and amplitude of the LOS channel signal components are random variables due to the statistical properties of the signal propagation medium.

Of current and practical interest is the consideration of the problem of developing a strategy for efficient coordinate estimation of the hybrid RSS/TDoA method for UWB Sensors Networks Using Two Anchor Nodes under conditions when the First Detected Peak of the AN receiver coincides with the DP channel component and the influence of the NLOS channel on the LOS channel can be neglected. It is assumed that the statistical properties of the LOS channel signal are described by a log-normal distribution.

Some results of this study were reported by the authors at a conference [46].

The structure of the paper is as follows. Section 2 of the article considers a description of the mathematical model of signals in TDOA/RSS location systems, which are analyzed in this paper. A mathematical model of a UWB radio pulse—a Gaussian monocycle is presented, as well as the properties of the probability density of the logarithmically normal distribution of the signal amplitude during its propagation in a medium. Section 3 describes the Hybrid TDoA/RSS Estimator of propagation delay time and signal amplitude. Section 4 provides a derivation of the Cramer–Rao lower bound for the delay time and signal amplitude estimates. A description of the source distance estimator and its statistical characteristics are also given. Section 5 presents the results of a computational experiment for estimating the parameters. Finally, the concluding remarks and future works are presented in Section 6.

## 2. Problem Statement and System

Let us define the class of TDOA/RSS location systems, the analysis of which is carried out in this paper. Consider two base stations of the location system, one of which is the reference station. ANs (sensors) operate in the corporate synchronous mode of common time *t*, and their position is given by known coordinates (*x_i_*, *y_i_*) *i* = 1, 2, …, *P*, where *P* is the AN number in the location system (see Figure 1). ANs receive a radio signal *s*(*t*′) from a fixed-point source (S) with unknown coordinates (*x*, *y*) located at a distance *r_i_* from the corresponding *i*-th base station (see Figure 1). We assume that the source emits a signal of a known function of time *s*_0_(*t*′), the reference points of the time axes *t* and *t*′ are shifted by an unknown value δ*t*, and the operation of the source transmitter is asynchronous in time with the operation of the BS receivers. The mathematical model of received by AN signals *x_i_*(*t*) in discrete time is described by the following relationships: (2)$$s_{0}={\left[s_{0}\left(t_{1}^{\prime }\right),s_{0}\left({t^{\prime }}_{2}\right),\ldots s_{0}\left(t_{N}^{\prime }\right)\right]}^{T},$$
(3)$$s_{i}={\left[s_{i}\left(t_{1}^{\prime }-\tau_{i}\right),s_{i}\left({t^{\prime }}_{2}-\tau_{i}\right),\ldots s_{i}\left({t^{\prime }}_{N}-\tau_{i}\right)\right]}^{T},$$
(4)$$n_{i}={\left[n_{i}\left(t_{1}^{\prime }\right),n_{i}\left(t_{2}^{\prime }\right),\ldots n_{i}\left(t_{N}^{\prime }\right)\right]}^{T},$$
(5)$$x_{i}={\left[x_{i}\left(t_{1}^{\prime }\right),x_{i}\left(t_{2}^{\prime }\right),\ldots x_{i}\left({t^{\prime }}_{N}\right)\right]}^{T},$$
(6)$$x_{i}=A_{i}s_{i}+n_{i}\;,$$
where ***s*_0_** is the source signal S; ***s_i_*** is delayed by τ*_i_* and attenuated signal of source S; ***x_i_*** is input signal of the *i*-th sensor receiver; *n_i_* is white Gaussian noise reduced to the input of the *i*-th sensor, uncorrelated with the source signal; and the symbol *T* means the vector transposition operation. 

The sampling interval Δ*t* = *t_k_*_+1_ − *t_k_* is determined by the operating frequency range Δ*f* of the emitted signal *s(t)*, with Δ*t* = 1/Δ*f*. Thus, at the observation time interval *T*_0_ of the signal *x_i_*(*t*), the number of uncorrelated samples of the sample group and, accordingly, the dimension of the vectors is 1 × *N*, where *N = T*_0_/Δ*t* = *T*_0_Δ*f*. The probability density distribution of noise counts *n_i_*(*t_k_*) = *n_k_* is considered to be given *p_n_*(*n_k_*) = ℵ(0,σ^2^*_n_*), i.e., noise samples *n_k_* are statistically independent centered Gaussian random variables. Such a distribution class adequately describes the model of the impact of quasi-stationary noise inherent in the receiver itself, as well as other additive processes such as thermal noise at the receiver input. As a rule, the use of non-Gaussian noise models, for example, impulsive random processes, does not allow one to obtain a closed analytical form of optimal signal processing algorithms. This disadvantage makes it problematic to conduct transparent analysis when using very powerful statistical tools, such as maximum likelihood estimation and Bayesian estimation [47,48]. However, in some media, for example, for the ocean noise model, the use of non-Gaussian distributions is necessary [47].

The amplitude *A_i_* of the wanted signal at the input of the *i*-th sensor is determined by the power of the source transmitter S and the law of attenuation of the electromagnetic wave in the location space. As noted above, we will take into account the statistical properties of the signal propagation medium in the approximation, when the influence of NLOS channels on LOS channels is small and the equation for the squared random value of the signal amplitude (power) is as follows:(7)10lgAi2A02=−10γ·lgrir0+nP,
where *n*_P_−is a centered Gaussian random variable with *p*_P_(*n*_P_) = *ℵ*(0,σ^2^_P_) probability density distribution. The reference power value *A*_0_^2^ in relation (7) is measured, for example, at a distance *r*_0_ = 1 m from the source. In works [17,23,24], the values of coefficients σ^2^_P_ and γ of distribution (7) are given for different conditions of source signal propagation. The standard deviation (RMS) σ_P_ can reach 12 dB and the γ parameter ranges from 2 (free space) to 6 (for multipath propagation in a multicomponent environment, for example, indoors).

Relation (7) determines the log-normal distribution of the probability density of the amplitude *A*_i_ [49] with an average value M{*A*_i_}
(8)$$M\left\{A_{i}\right\}={\;A}_{0}{\left(\frac{r_{0}}{r_{i}}\right)}^{\frac{\gamma }{2}}\exp\left[\frac{1}{2}\frac{{\left(\sigma_{P}\right)}^{2}{\left(\ln10\right)}^{2}}{20}\right]=A_{0}{\left(\frac{r_{0}}{r_{i}}\right)}^{\frac{\gamma }{2}}\exp\left(\frac{1}{2}\sigma_{\mathrm{Pe}}^{2}\right),$$
and variance D{*A*_i_} equal to [49]
(9)$$D\left\{A_{i}\right\}={\;\sigma }_{A_{i}}^{2}={\;A}_{0}^{2}{\left(\frac{r_{0}}{r_{i}}\right)}^{\gamma }{\exp(\sigma }_{\mathrm{Pe}}^{2})\left[\exp\left(\sigma_{\mathrm{Pe}}^{2}\right)-\;1\right],$$

For the case when the variance σ^2^_Pe_ of the normalized signal power is much less than one, the asymmetry and kurtosis coefficients of the probability density distribution of the parameter *A*_i_ are equal to a small value, and, therefore, the distribution law of the probability density of the amplitude *p*(*A*_i_) can be considered approximately normal [49].
(10)$$p\left(A_{i}\right)=\frac{1}{\sqrt{2{\pi \sigma }_{A_{i}}^{2}}}\exp\left[-\frac{1}{2}\frac{\left(A_{i}-M\left\{A_{i}\right\}\right)}{\sigma_{A_{i}}^{2}}\right],$$

Figure 2a shows the change in the random value of the amplitude *A*_i_ for various values of the parameter σ*_P_* = 6 dB and 1.5 dB of the lognormal distribution. In the proposed model for describing the channel for the propagation of microwave signals in a medium, it is assumed that during the time interval when the amplitude *A*_i_ and the delay time τ are estimated, the values of these parameters remain unchanged. In the next measurement session, with the same coordinates of the source SN, the amplitude *A*_i_ has a random value from the general population with a lognormal distribution. In Figure 2b, the probability distribution density (PDF) of the amplitude (b) for various values of the parameter σ*_P_* of the lognormal distribution is shown. For σ_P_ variances less than 1.5 dB, the lognormal and normal (ND) distributions almost coincide. For large values of dispersion, the probabilistic interval of changes in the amplitude *A*_i_ with a lognormal distribution significantly exceeds the corresponding interval with a normal distribution of a random variable. This feature is explained by the flat “tails” of the lognormal distribution (see Figure 2b). Another feature of the lognormal distribution in accordance with relations (8) and (9) is the dependence of M{*A*_i_} and D{*A*_i_}, simultaneously, on two distribution parameters σ^2^_Pe_ and μ*_i._*
(11)$$\mu_{i}=\gamma \cdot \mathrm{lg}\left(\frac{r_{0}}{r_{i}}\right),$$

The impulse response of the channel is affected by the polarization of the radio wave and the radiation pattern of the receiving and emitted antennas. In this paper, these effects are not studied. It should also be noted that although the Saleh–Valenzuela model was developed for None Line of Sight (NLOS) channels, it has also been applied to Line of Sight (LOS) channels, where it is more limited [23].

As noted above, for the tasks of radar and navigation with short ranges and high spatial resolution, a promising direction is the use of UWB technologies. The main definitions and technical features of UWB technologies are given in [50,51]. To analyze the RSS/TDoA-Based Source Localization hybrid method being developed, we chose an ultra-short radio pulse as a Waveform UWB signal, for which the change in the envelope over time is described by the first derivative of the Gaussian function. The waveform of the envelope of the UWB signal and the module of the amplitude spectrum are shown in Figure 3. The envelope of the emitted signal *s*_0_(*t*) as a known function of time is described by the expression
(12)$$s_{0}(t)\;=A_{0}\frac{t}{\tau_{0}}\exp\left[-{\left(\frac{t}{\tau_{0}}\right)}^{2}\right]\mathrm{rect}\left[\frac{t}{T}-\frac{1}{2}\right]=A_{0}s\left(t\right),$$

In Formula (12), *A*_0_ is the signal amplitude at the reference distance *r*_0_, which determines the pulse power of the emitted signal *P*_0_ ~ *A*^2^_0_/4; τ is the parameter of the UWB pulse and *T* is effective pulse width. Not less than 99% is concentrated on the time interval *T* at *T* = 4τ_0_. The rectangular function rect(x) (unit pulse) is expressed in a standard way in terms of the Heaviside function [52]. Figure 2a also shows the ‘mother’Gaussian function (2), the derivative of which generates the shape of the envelope of the emitted signal.

Modern receiver modules use the technology of heterodyning and the IQ quadrature conversion of the input signal. In further analysis, it is assumed that the radio signal with the center frequency *f*_0_ is converted into a signal during heterodyning, of which the mathematical model is described by relation (12). We also assume that the medium within the direct path of signal propagation is weakly dispersive in the frequency band of the operating range, and the distortions of the shape of the envelope (12) are small. It is assumed that the influence of the NLOS channels on the LOS channels is small due to the choice of a small parameter τ and, accordingly, the small duration of the interval *T* compared to the additional time delay in the NLOS channels. The introduced restrictions make it possible to develop an optimal algorithm for processing the received signal in the joint TDoA/RSS assessment mode.

## 3. The Proposed Hybrid TDoA/RSS Estimator

For further analysis, it is advisable in the mathematical model of the signal (1)–(5) to move from the space of discrete time to the space of discrete frequency ω_k_. The signal spectrum sampling interval δ*f* = *f_k_*_+1_ – *f_k_* is equal to δ*f =* Δ*f/N*, then ω_k_ = 2π*f_k_* = 2π*k*δ*f*. After performing the DFT on the received signal *x*(*t*), we obtain a complex-valued vector of the signal spectrum.
(13)$$\tilde{X_{i}}={[\tilde{X_{i}}{(f}_{1}),\;\tilde{X_{i}}{(f}_{2}),\;\ldots \tilde{X_{i}}{(f}_{N})]}^{T}\;,$$

Similar expressions for the discrete spectrum can be written by performing the DFT on the signal *s*(*t*) and the noise *n*(*t*). Then, the mathematical model of the received signal can be represented as follows:(14)$$\tilde{X_{i}}={\;A}_{i}{\tilde{S}}_{i}e^{-{j\omega }_{k}\tau_{i}}+\tilde{N_{i}},$$

In Equation (13), the vector ***X*** has a multivariate complex normal distribution with mean ***m***(**θ**) and covariance matrix **K** [53,54,55]
(15)$$Cℵ(\tilde{m_{i}}\left(\theta \right),\;K),$$
and mathematical expectation.
(16)$$\tilde{m_{i}}(\theta )\;={\;A}_{i}{\tilde{S}}_{i}e^{-{j\omega }_{k}\tau_{i}},$$

The covariance matrix **K**, in our case of statistically independent elements of the vector ***X*_i_**, is expressed through the identity matrix.
(17)$$K=\frac{N_{0}}{2}T_{0}\cdot \overset{=}{1},$$
where N_0_/2 is two-sided noise power spectral density *n*(*t*), and **θ** = [*A*_i_, τ_i_]^T^ is the vector of parameters to be estimated.

In most radio engineering problems, when estimating signal parameters, the maximum a posteriori density (MAPD) method (or the maximum a posteriori (MAP) estimate) is mainly used [53,54]. The estimates obtained by the MAPD have a number of advantages—the comparative simplicity of calculations and the technical implementation of the corresponding optimal algorithms using modern digital processors. An important advantage of the MAP estimate is that the estimate is asymptotically efficient for a large sample size ***X***_i_ [53,54].

For a complex vector sample ***X***_i_ of statistically independent samples, the joint conditional probability distribution density *p*_X_ (***X***_i_|**θ**) has the form [53,55].
(18)$$p\left(X_{i}|\theta \right)=\frac{1}{\det[\pi K]}\exp\left\{-{\left[{\tilde{X}}_{i}-\tilde{m}(\theta )\right]}^{H}\left[{}_{2}^{1}K^{-1}\right]\left[{\tilde{X}}_{i}-\tilde{m}(\theta )\right]\right\},$$
where **K**^−1^—is the inverse matrix of the covariance matrix **K**. This is according to relation (15), in our case.
(19)$$K^{-1}=\frac{1}{T_{0}N_{0}}\cdot \overset{=}{1},$$

Bayesian MAP estimation seeks the maximum of the Bayesian log-likelihood function (log-likelihood function) [53,54].
(20)$$l_{B}\left(\theta ,{\tilde{X}}_{i}\right)=\ln p\left({\tilde{X}}_{i}|\theta \right)+\ln p_{\theta }(\theta )+\zeta ,$$

In our case, the form *p*_θ_(**θ**) = *p*_A_(*A*_i_) is determined by the relation for the lognormal distribution LogN(μ_i_,σ^2^_P_) [49].
(21)$$p_{A}(A_{i})=\mathrm{LogN}(i{,\sigma }_{P}^{2})=\frac{M_{1}}{A_{i}\sigma_{P}\sqrt{2\pi }}\exp\left[-\frac{1}{2}\frac{{\left(\log A_{i}\;-\;\mu_{i}\right)}^{2}}{\sigma_{P}^{2}}\right]\;for\;A_{i}>0,$$

Parameter *M*_1_= lg(e); ζ—is a function independent of the vector parameter **θ**. In expression (21), for the Bayesian log-likelihood function *l*_B_(**θ**,***X***), the first term takes into account the information obtained from the observation results and the second—a priori information.

The likelihood equation that determines the estimate of the vector parameter **θ** can be written in a compact form [53,54].
(22)$$\nabla_{\theta }{\left[l_{B}\left(\theta ,{\tilde{X}}_{i}\right)\right]}_{\theta =\hat{\theta }(\tilde{X})},$$
where ∇_θ_ is the vector symbol of a multidimensional 2 × 1 gradient.
(23)$$\nabla_{\theta }\;≜{\left[\frac{\partial }{\partial A_{i}}\;,\;\frac{\partial }{\partial_{i}}\right]}^{T}\;\;,$$

A more general algorithm for estimating the vector parameter **θ** has the form [53,54].
(24)$$\theta (\tilde{X})=\underset{\theta }{argmax}\left[l_{B}\left(\theta ,{\tilde{X}}_{i}\right)\right],$$

Using the results obtained in [53,54], we obtain the following expressions for estimating the vector parameter **θ.**
(25)$$ℜ{\left\{\frac{\partial {\tilde{m}}^{H}(\theta )}{\partial A_{i}}K^{-1}\left[{\tilde{X}}_{i}-\tilde{m}[\theta ]\right]\right\}}_{\theta =\hat{\theta }}+{\frac{\partial p_{A}\left(A_{i}\right)}{\partial A_{i}}}_{\theta =\hat{\theta }}=0,$$
(26)$$ℜ{\left\{\frac{\partial {\tilde{m}}^{H}(\theta )}{\partial \tau_{i}}K^{-1}\left[{\tilde{X}}_{i}-\tilde{m}[\theta ]\right]\right\}}_{\theta =\hat{\theta }}=0$$
where ℜ{⋅} denotes taking the real part of a complex-valued function. To obtain a solution in a closed form, in practically all important cases, we consider that the solution of Equation (25) is determined by the first term for any given non-random parameter μ_i_ from the spatial region of the radiation source.

The obtained relations allow, after carrying out the corresponding calculations, to obtain the following optimal asymptotically efficient estimates of the signal amplitude *A*_i_ and the delay time τ_i_.
(27)$${\hat{\tau }}_{i}=\underset{\tau }{\mathrm{argmax}}\left\{\overset{N}{\underset{k=1}{\sum }}x_{i}\left(t_{k}\right)s\left(\tau +t_{k}\right)\right\}$$
(28)$${\hat{A}}_{i}=\left(ℜ\left\{{\left[\tilde{S}e^{-j\omega {\hat{\tau }}_{i}}\right]}^{H}{\tilde{X}}_{i}\right\}\right)/{\tilde{S}}^{H}\tilde{S}=\sum_{k=1}^{N}x_{i}\left(t_{k}\right)s\left(\hat{\tau }+t_{k}\right)/\sum_{k=1}^{N}s^{2}{\left(t_{k}\right)}^{\prime }$$

The optimal estimate of the signal delay time τ in accordance with relation (27) indicates that the value of the estimate τ coincides with the maximum of the correlation integral of the input signal *x*(*t*) and the signal *s*(*t* + τ) with a variable time shift τ. When calculating the optimal amplitude estimate, the value of the delay time τ is considered to be a given value.

It is easy to show that the obtained estimates of the delay time τ and the signal amplitude *A*_i_, as follows from the properties of the maximum a posteriori (MAP) estimate method [53,54], are unbiased and
(29)$$M\left\{{\hat{A}}_{i}\right\}=A_{i}$$

## 4. CRLB for Estimating Delay Time and Signal Amplitude. Source Range Estimator

The root-mean-square errors (RMS) of the obtained optimal estimates of the parameters τ_i_ and *A*_i_ satisfy the Bayesian Cramer–Rao lower bound (BCRLB), and, for the sample size of the observed signal *N*→∞, BCRLB coincides with the RMS derivation due to the asymptotic efficiency of the MAP parameter estimation method. The error covariance matrix for estimating the parameters τ_i_ and *A*_i_ is determined by the Bayesian Information Matrix (BIM) **J**_B_ with a size of 2 × 2 elements [53,54].
(30)$$J_{B}(\theta )=-M\left\{\nabla_{\theta }{\left[\nabla_{\theta }l_{B}\left(\theta ,{\tilde{X}}_{i}\right)\right]}^{T}\right\}=M\left\{\left[\nabla_{\theta }l_{B}\left(\theta ,{\tilde{X}}_{i}\right)\right]{\left[\nabla_{\theta }l_{B}\left(\theta ,{\tilde{X}}_{i}\right)\right]}^{T}\right\},$$
where the Bayesian log-likelihood function *l*_B_ is defined by expression (20).

The calculation of the Bayesian lower bound of the covariance matrix of estimates **C**_ε_ is based on the Cramer–Rao inequality.
(31)$$C_{\varepsilon }\;≜\;M\left\{{\left[\hat{\theta }-\theta \right]}^{H}\left[\hat{\theta }-\theta \right]\right\}\geq J_{B}^{-1}\left(\theta \right),$$
where **J**_θ_^−1^(**θ**) — is the matrix inverse to the Bayesian information matrix (29). The diagonal elements in the matrix (30) **C**_τ_ and **C***_A_* are equal to the lower Bayesian bound of the corresponding mean square errors of the parameter estimates, so the off-diagonal BIM elements **C**_τ,*A*_ determine the error correlation.

With the mathematical model is used for describing the location system, the calculation of BIM leads to the following result:(32)$${\left[J_{B}^{-1}\left(\theta \right)\right]}_{\tau ,A}={\left[J_{B}^{-1}\left(\theta \right)\right]}_{A,\tau }=\;0,$$
i.e., the mutual errors in the estimates of the amplitude *A*_i_ and the signal delay time τ_I_ are not correlated.

The formulas for calculating the diagonal elements of the matrix **C**_ε_ for signals discrete in time have the form
(33)$$C_{A}=\sigma_{a}^{2}={\left[\frac{\sum_{j}{\left|\tilde{S}\left(f_{j}\right)\right|}^{2}}{\frac{1}{2}N_{0}T_{0}}+\frac{1}{\sigma_{A_{i}}^{2}}\right]}^{-1}={\left[\frac{\sum_{j}{\left|s\left(t_{j}\right)\right|}^{2}}{\sigma_{n}^{2}}+\frac{1}{\sigma_{A_{i}}^{2}}\right]}^{-1},$$
(34)$$C_{\tau }=\sigma_{\tau }^{2}=\frac{\sigma_{n}^{2}}{A_{i}^{2}\sum_{j}{\left|\frac{\partial s_{i}\left(t_{j}\right)}{\partial t}\right|}^{2}},\;$$
where, as before, the parameters σ*_n_* and σ*_A_* mean the magnitude of the receiver noise dispersion over the entire operating frequency range and the dispersion of the signal amplitude in the lognormal distribution (9) in the Gaussian approximation (10).

Consider the implementation of the algorithm for optimal estimation of distances *r*_1_ and *r*_2_ from the radiation source SN to the corresponding two base stations AN_1_ and AN_2_ (see Figure 1). Estimated parameters *r*_1_ and *r*_2_ are unknown and non-random values. During one measurement session, in accordance with the proposed algorithm (27), (28), the optimal asymptotically efficient unbiased estimates of the signal amplitudes *A*_1_, *A*_2_, and the delay time difference τ_12_. For the mathematical model of the measurement results, the vector ***R*** ≡ [*R*_1_, *R*_2_, *R*_3_]^T^ can be represented as:(35)$$R_{1}=v_{g}\left(\;{\hat{\tau }}_{1}-\;{\hat{\tau }}_{2}\right)=v_{g}\;{\hat{\tau }}_{12}=\left(r_{1}-r_{2}\right)+n_{1},\;$$
(36)$$R_{2}={\hat{A}}_{1}=A_{1}+n_{2},\;$$
(37)$$R_{3}={\hat{A}}_{2}=A_{2}+n_{3},\;$$
where v_g_ is the group velocity of signal propagation in the medium, and the measurement noise *n*_1_, *n*_2_, and *n*_3_ are centered random variables having normal probability distribution densities *w*_1_, *w*_2_, and *w*_3_ with parameters.
(38)$$w_{1}=ℵ\left(0,v_{g}^{2}\sigma_{\tau }^{2}\right);\;\;\;\;w_{2}=w_{3}=ℵ\left(0,\sigma_{0}^{2}\right),\;$$

The variance σ_τ_^2^ is determined by relation (33), and the variance σ_0_^2^ is determined by the first term in square brackets (32).

Random amplitudes *A*_1_ and *A*_2_ in the expressions have a log-normal distribution with parameters μ_1_, μ_2_ and σ^2^_P_ in accordance with (21). The proposed model allows us to determine the probability distribution densities *p*_1_, *p*_2_, and *p*_3_ of the measurement results *R*_1_, *R*_2_, and *R*_3_:(39)$$p_{1}\left(R_{1}|r_{1}-r_{2}\right)=\frac{1}{v_{g}\sigma_{\tau }\sqrt{2\pi }}\exp\left[-\frac{1}{2v_{g}^{2}\sigma_{\tau }^{2}}{\left(R_{1}-r_{1}+r_{2}\right)}^{2}\right],$$
(40)$$p_{i}\left(R_{i}|r_{1},r_{2}\right)=\int_{-\infty }^{\infty }\frac{M_{1}}{A_{i}\sigma_{P}\sqrt{2\pi }}\exp\left[-\frac{1}{2}\frac{{\left(lgA_{i}{-}_{i}\right)}^{2}}{\sigma_{P}^{2}}\right]\frac{1}{\sigma_{0}\sqrt{2\pi }}\exp\left[-\frac{1}{2\sigma_{0}^{2}}{\left(R_{i}-A_{i}\right)}^{2}\right]dA_{i},\;i=2,3,$$

The probability distribution densities *p*_2_ and *p*_3_ in (39) are calculated through the convolution integral of the one-dimensional densities w_2,3_ and *p_A_*(*A*_i_) [56]. In practical applications, the inequality σ_P_ >> σ_0_ is usually satisfied. In this case, the probability distribution densities *p*_2_ and *p*_3_ approximately have a lognormal law (21).

To estimate the radii *r*_1_ and *r*_2_ as unknown but nonrandom parameters, we use the Maximum Likelihood (ML) strategy. This strategy is based on finding the maximum of the logarithm of the likelihood function ln *l*(***R***, *r*_1,_ *r*_2_) and solving the likelihood equation [53,54] for the two-dimensional case. Two scenarios are possible when developing a range estimator to a radiation source.

The first scenario—the transmitter power, and, hence, the amplitude of the signal *A*_0_ at the reference point, are considered a priori given values. In this case, the estimate of the radii *r*_1_ and *r*_2_ follows from the solution of the system of equations:(41)$$\frac{\partial \ln l(R,r_{1},\;r_{2}\;)}{\partial r_{i}}=0,\;\;\;\;\;\;i=1,2,\;$$
(42)$$\ln l(R,r_{1},\;r_{2}\;)=\ln p_{1}\left(R_{1}|r_{1}-r_{2}\right)+\ln p_{2}\left(R_{2}|r_{1},r_{2}\right)+\ln p_{3}\left(R_{3}|r_{1},r_{2}\right),\;$$

The solution of the transcendental Equation (41) for the unknowns *r*_1_ and *r*_2_ can be carried out numerically or approximately by analytical methods. The analysis performed shows that due to the determination of two unknowns *r*_1_ and *r*_2_ from the results of three measurements *R*_1_, *R*_2_, and *R*_3_, the accuracy of determining the coordinates SN of the radiation source increases.

Let us consider in more detail the scenario when the transmitter power is not known a priori. This case is more common in practice, since the signal amplitude *A*_0_ at the reference point depends on the radiation pattern of the receiver and transmitter antennas, technological deviations from the average values of the parameters of the devices that make up the network, and so on. In this case, it is advisable to choose the ratio *R*_1_/ *R*_2_ as the result of the second measurement *Y*. Due to the fact that the random variables *R*_1_ and *R*_2_ have probability distribution densities close to log-normal, their ratio also has a log-normal distribution [49].
(43)$$p_{Y}\left(Y|r_{1},r_{2}\right)=p_{Y}\left(\frac{R_{1}}{R_{2}}|r_{1},r_{2}\right)=\frac{M_{1}}{Y\sigma_{P}\sqrt{4\pi }}\exp\left[-\frac{1}{2}\frac{{\left(\mathrm{lg}Y-\frac{\gamma }{2}\mathrm{lg}\frac{r_{2}}{r_{1}}\right)}^{2}}{2\sigma_{P}^{2}}\right],$$

The estimation of the parameters *r*_1_ and *r*_2_ follows from the solution of the system of equations:(44)$$\frac{\partial \ln l(R_{1},Y,\;r_{1},\;r_{2}\;)}{\partial r_{1}}=0,\;\;\;\;\;\;\frac{\partial \ln l(R_{1},Y,\;r_{1},\;r_{2}\;)}{\partial r_{2}}=0,\;$$
(45)$$\ln l(R_{1},Y,\;r_{1},\;r_{2}\;)=\ln p_{1}\left(R_{1}|r_{1}-r_{2}\right)+p_{Y}\left(Y|r_{1},r_{2}\right),\;$$

The solution of the system of Equation (43) in the final (closed) form is possible when the value of the indicator γ of the signal attenuation during its propagation in the medium is equal to two. In this case
(46)$$r_{1}=\frac{R_{1}}{1-Y};\;\;\;\;\;\;\;\;\;r_{2}=\frac{R_{1}Y}{1-Y}\;,\;$$

Using the rule of approximate calculation of the mathematical expectation of a random variable, it is easy to obtain the statistical characteristics of the estimation of the parameters *r*_1_ and *r*_2_ [56].
(47)$$M\left\{r_{1}\right\}\approx \frac{M\left\{R_{1}\right\}}{1-M\left\{Y\right\}}=\frac{r_{1}-r_{2}}{1-\frac{r_{2}}{r_{1}}\alpha },\;$$
(48)$$M\left\{r_{2}\right\}\approx \frac{M\left\{R_{1}\right\}\cdot M\left\{Y\right\}}{1-M\left\{Y\right\}}=\frac{\left(r_{1}-r_{2}\right)\frac{r_{2}}{r_{1}}\alpha }{1-\frac{r_{2}}{r_{1}}\alpha },\;$$

In expressions (46) and (47), the parameter α is determined by the following relation:(49)$$\alpha =\exp\left(\sigma_{\mathrm{Pe}}^{2}\right),$$

It follows from relations (46) and (47) that as the variance σ^2^_P_ tends to zero, the estimates of the parameters *r*_1_ and *r*_2_ are unbiased. With a finite value of the dispersion σ^2^_P_, these estimates become biased.

Similarly, using the rule of approximate calculation of the variance of a random variable, it is easy to obtain the variance of the estimates of the parameters *r*_1_ and *r*_2_ [56].
(50)$$\frac{D\left\{r_{1}\right\}}{r_{1}^{2}}=\frac{1}{{\left(1-\alpha \frac{r_{2}}{r_{1}}\right)}^{2}}\left[\frac{2\sigma_{\tau }^{2}v_{g}^{2}}{r_{1}^{2}}+\frac{{\left(1-\frac{r_{2}}{r_{1}}\right)}^{2}{\left(\frac{r_{2}}{r_{1}}\right)}^{2}}{{\left(1-\alpha \frac{r_{2}}{r_{1}}\right)}^{2}}\alpha^{2}\left(\alpha^{2}-1\right)\right],\;$$
(51)$$\frac{D\left\{r_{2}\right\}}{r_{2}^{2}}=\frac{1}{{\left(1-\alpha \frac{r_{2}}{r_{1}}\right)}^{2}}\left[\frac{2\sigma_{\tau }^{2}v_{g}^{2}}{r_{1}^{2}}+\frac{{\left(1-\frac{r_{2}}{r_{1}}\right)}^{2}}{{\left(1-\alpha \frac{r_{2}}{r_{1}}\right)}^{2}}\alpha^{2}\left(\alpha^{2}-1\right)\right],\;$$

Estimation of the parameters *r*_1_ and *r*_2_ in the general case does not lead to an unambiguous determination of the coordinates of the radiation source SN, since the radii *r*_1_ and *r*_2_ intersect at two spatial points symmetrically with respect to the base—the line connecting the two base stations AN_1_ and AN_2_ (see Figure 1). To unambiguously determine the SN coordinates, it is necessary that all sources are on the same side of the base.

## 5. Simulation Results and Discussion

To check the obtained theoretical results and the adequacy of the proposed mathematical models, a computer simulation (machine experiment) was carried out, during which the statistical characteristics of the obtained parameter estimates were determined. When performing the computer simulation, the selection of the numerical values of the parameters UWB of the Gaussian monocycle radio pulse, the time sampling frequency *f*_s_, the number of samples in the signal sample *N*, and other parameters was based on the resource of modern and promising hardware as well as the corresponding technical implementations based on them.

The power of the SN transmitter was determined by the requirements of the USA Federal Communications Commission (FCC) and was minus 17 dBm during the simulation. This parameter corresponds to a power spectral density level of about minus 50 dBm/MHz in the frequency band from 3 to 10 GHz. The parameter τ of the UWB pulse was chosen to be 1.15 ns, which determined the effective operating bandwidth of the signal Δ*f*_e_ equal to about 500 MHz. The number of samples in the signal sample *N* was chosen to be 1024 and 2048. This choice is dictated by the size of the memory buffer of modern high-speed processors. In addition, the value of *N* determines the sampling duration *T*_0_ = *N*/*f*_s_, and, consequently, the maximum range of MSNs for the *R*_max_ at a given sampling rate. In modeling, the frequency *f*_s_ ≥ 2Δ*f*_e_, which determines the time sampling step Δ*t* = 1/*f*_s_, was chosen to be 3 GHz and 6 GHz, which corresponded to 24 and 48 counts at the effective duration of a Gaussian unicycle. The value of *r*_1_ in the simulation was chosen to be 33 m. To obtain the sample mean and sample variance of the proposed estimates, multiple repetitions of computer calculations were carried out for different implementations of noise, followed by statistical processing.

a shows the implementation of the input signal *x(t)* with an SNR of 12 dB. Figure 4b shows the result of processing the signal *x*(*t*), which implements a point estimate of the signal arrival time τ_t_ by the maximum of the function *R*_x_(τ) defined by Formula (27). The *R*_x_(τ) function as a random process has fluctuations (see Figure 4b), the level of which increases as the input noise increases (SNR decreases). When statistically processing a large number of a set of realizations, the probability *P* of a random event that the level of one individual sample *R*_x_(τ) of the *n*-th realization will exceed the maximum value, and the signal arrival time will be estimated in the sampling interval Δ*t*_k_ anomalously far from τ_t_, as determined by the formula
(52)$$P=\mathrm{erf}\frac{\beta }{\sqrt{2}}\;,\;$$
where erf(.) is the error function. In this case, the variance of the estimate increases sharply. When the value of the parameter β is less than 4.6, the probability of such an error is low. This condition imposes a restriction on the lower limit of the change in the SNR parameter for a point estimate of the signal arrival time, which turns out to be approximately 5 dB. With a smaller SNR value, it is necessary to use the methods of interval parameter estimation. The upper limit of the SNR parameter for a given transmitter power SN is determined by the distance to AN and the minimum possible noise figure of the receiver, which was assumed to be several decibels in the simulation in the absence of a cryogenic cooling system.

Figure 5 shows a comparison of the dependences of the normalized standard deviations of the delay time estimate and the signal amplitude estimate on the SNR value for various values of the sample size *N* = 1024 and *N* = 2048. The blue color shows the graphs of the results of calculating the parameter estimates using the proposed algorithm (relations (27) and (28)), while in red are the plots of the Cramer–Rao lower bound of the corresponding estimate. As follows from the theoretical analysis, the estimate weakly depends on the sample size at a fixed value of the sampling rate *f*_s_ equal to 3 GHz and the UWB pulse parameter τ equal to 1.15 ns. The difference between the estimate of the signal arrival time τ_t_ and CRLB can be explained by the discrete nature of signal processing. A slight increase in the discrepancy between the obtained estimate and CRLB at *N* = 2048 is explained by an increase in the statistical error in determining sample estimates with an increase in the sample size.

Figure 6 shows a comparison of the dependences of the normalized standard deviations of the delay time estimate and the signal amplitude estimate on the SNR value for various values of the parameter τ_0_ UWB of the Gaussian unicycle radio pulse equal to 1.15 ns and 2.25 ns. The sampling rate in the computer experiment is 6 GHz and the sample size is *N* = 2048. The blue graphs display the results of the parameters estimation calculation according to the proposed algorithm, while the graphs of the Cramer–Rao Lower Bound of the corresponding estimate are displayed in red. The better coincidence between the arrival time estimate and CRLB compared to the previous case is explained by the doubling of the number of readings in the Gaussian unicycle duration interval. The closeness of the proposed estimate to CRLB, obtained as a result of the simulation, confirms the conclusion about the asymptotic efficiency of the estimate obtained using the MAP strategy.

Figure 7 shows the dependences of the normalized value of the average value of the estimate *r*_1_ on the parameter σ^2^_P_ and the two values of the ratio of radii. The estimate was normalized to the true value of the radius *r*_1_, the value of the parameter τ_0_ is 1.15 ns, the sampling frequency is 6 GHz, and the sample size is *N* = 2048. As the ratio *r*_2_/*r*_1_ approaches unity, the shift in the estimate of the average value of the radius increases sharply with the growth of the variance of the lognormal distribution σ^2^_P_. The point *r*_2_/*r*_1_ = 1 is singular in Equations (46)–(50). Thus, for the value σ^2^_P_ = 1 dB and the radius ratio *r*_2_/*r*_1_ = 0.5, the estimate shift reaches 40% (see Figure 7).

In Figure 8, dependences of the normalized standard deviation of the estimate *r*_1_ on the parameter σ^2^_P_ (a) and the SNR parameter (b) are presented. The estimate was normalized to the true value of the radius *r*_1_, the value of the parameter τ_0_ is 1.15 ns, the sampling frequency is 6 GHz, and the sample size is *N* = 2048. The dependence on the parameter σ^2^_P_ is given for two ratios of radii *r*_2_/*r*_1_ = 0.25 and *r*_2_/*r*_1_ = 0.5. The dependence on the SNR parameter is given for two values of the dispersion σ^2^_P_, the blue graph corresponds to the case σ^2^_P_ ≈ 0 and the red curve to the case σ^2^_P_ ≈ 0.3⋅10^−3^ dB, with the radius ratio *r*_2_/*r*_1_ = 0.2. As well as the bias of the radius estimate, the variance of the estimate increases sharply as the parameter *r*_2_/*r*_1_ approaches one. The simulation results show in this case, that when the variance of the lognormal distribution exceeds σ^2^_b_ = 10^−3^ dB, the main source of error in estimating the distance to AS is the statistical properties of the propagation medium of the UWB signal in the LOS channel.

Figure 9 shows 3D graphs of the statistical characteristics of the estimates of the radius *r*_1_, calculated by relations (46) and (49). The simulation parameters are chosen identically, in which the dependences presented in Figure 8 were calculated. As follows from relations (46) and (49) and, accordingly, from the graphs in Figure 9, the bias of the estimate (**a**) and the standard deviation of the estimate of the parameter *r*_1_ (**b**) begin to increase with increasing noise dispersion of the amplitude σ^2^_P_ in the logarithmic normal distribution (42), tending to be the ratio of radii *r*_2_/*r*_1_ to one. When deriving Formulas (45)–(50), it was assumed that the ratio *r*_2_/*r*_1_ is not equal to 1 one.

The statistical characteristics of the estimate of the radius *r*_2_ are similar to those for the radius *r*_2_. The obtained characteristics make it possible to determine the maximum value of the parameter σ^2^_P_ and the operating interval of the ratios *r*_2_/*r*_1_ by the given allowable values of the bias and standard deviation of the range estimate. It is also possible to solve the inverse problem by the given value of the parameter σ^2^_P_ and the working interval of the ratios *r*_2_/*r*_1_ to determine the values of the offset and the standard deviation of the range estimate.

## 6. Conclusions

The use of hybrid methods, such as RSS/TDoA, can improve the statistical characteristics of the spatial positioning of radio sources using Microwave Sensors Networks (MSN). One effective way to deal with multipath fading, including in small enclosed spaces, is to use UWB pulsed radio systems, due to their high time resolution capability. The work proposes a statistical model of a radio channel that adequately describes the propagation of a UWB signal under Line of Sight conditions and allows a closed-ended theoretical analysis and further computer simulation of the MSN statistical characteristics. For the first time, an algorithm for estimating the parameters of the spatial positioning of radio sources is proposed, taking into account a priori data on the logarithmically normal probability distribution density of a random value of the power UWB of a pulse signal, when it propagates in a medium with given statistical properties. The computer simulation performed showed that the dispersion of the obtained optimal estimate of the amplitude *A*_i_ and the signal propagation delay time τ based on the Bayesian strategy of maximum a posteriori probability in the operating range of the input noise level of the receiver are close to the Bayesian Cramer–Rao Lower Bound (BCRLB). For a scenario with an a priori unknown value of the transmitter power of the radiation source, an algorithm has been developed for optimal estimation of the distances from the source to the base stations based on the results of measurements of the amplitude and propagation delay time of the UWB radio signal. It is shown that when the variance of the log-normal distribution exceeds σ^2^_b_ (the value of σ^2^_b_ depends on the MSN parameters), the main source of error in estimating the distance to AS is the statistical properties of the propagation medium of the UWB signal in the LOS channel. The bias and variance of the obtained estimate of the distance from the source to the base stations increase with increasing distance, the noise level of the receiver, and the parameter σ_P_ of the of the log-normal distribution of the UWB signal power. An increase in the bias and variance of the obtained estimate is also observed when the value of the ratio of radii *r*_2_/*r*_1_ approaches a value equal to one. This evaluation behavior is related to the method of solving the system of equations and may be solved by a modified method in the future.

For further work, it is planned to refine the developed algorithm for estimating parameters in the presence of multiple radar targets and several base stations, as well as algorithms, for tracking the trajectory of unmanned vehicles with subsequent filtering algorithms. The mode of operation of an automotive radar in conditions of multiple radar targets and monitoring indoors of a limited volume is important in the practical use of the algorithm in modern driver assistance systems (ADAS—Advanced Driver Assistance System) to ensure the most comfortable and safe vehicle movement, including in the fully autonomous (unmanned) mode of vehicle movement. Moreover, for further work to validate the integrity of approaches to analyzing the performance of the developed algorithm, it is planned to develop an experimental setup of a hardware-software complex for the radio-frequency positioning of objects, based on research results [57] and completed comparative test measurements.

## Figures and Tables

**Figure 1 sensors-22-03018-f001:**
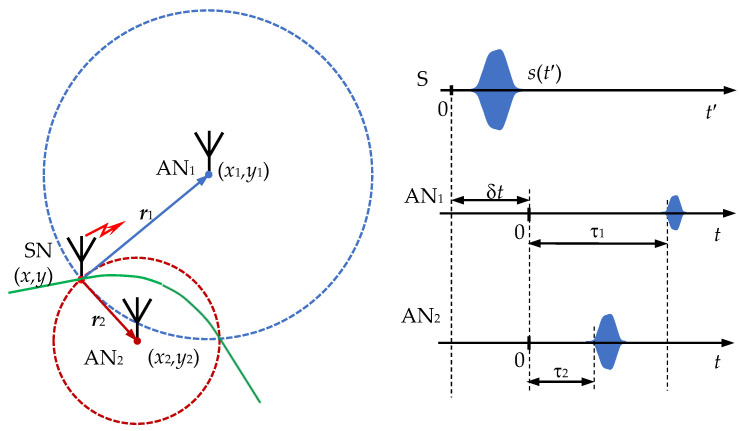
Block diagram of the TDoA/SSR-based positioning system.

**Figure 2 sensors-22-03018-f002:**
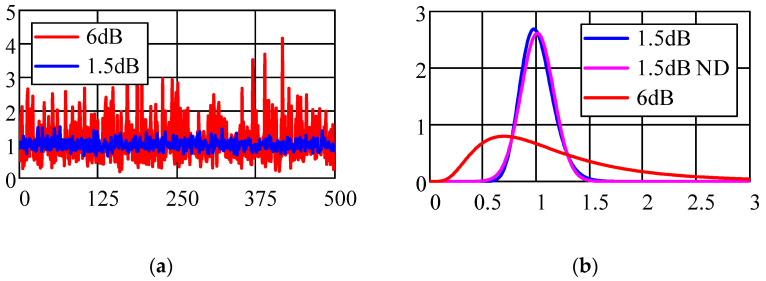
Change in the amplitude of the UWB signal (**a**) and the probability distribution density of the amplitude (**b**) for different values of the parameter σ_P_ of the lognormal distribution.

**Figure 3 sensors-22-03018-f003:**
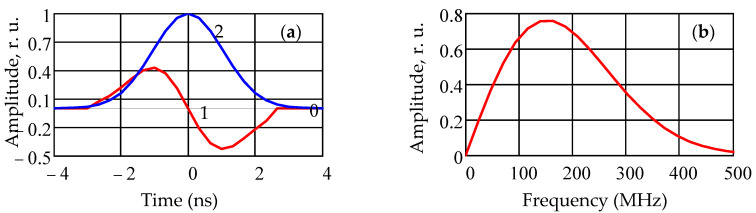
Waveform of the UWB signal (**a**) and the modulus of its amplitude spectrum (**b**).

**Figure 4 sensors-22-03018-f004:**
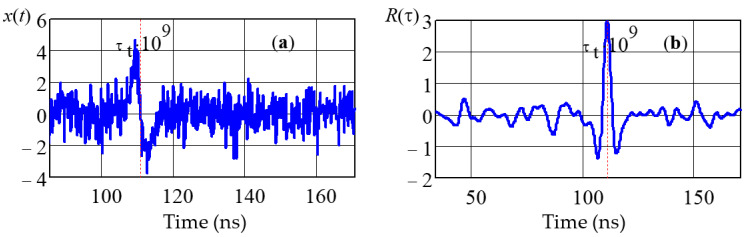
Implementation of the input signal *x*(*t*) (**a**) and the output signal of the estimator (**b**).

**Figure 5 sensors-22-03018-f005:**
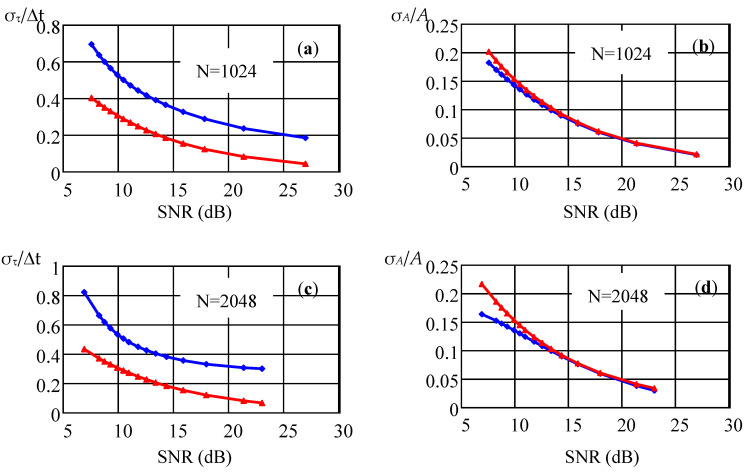
Dependences of the normalized standard deviations of the delay time estimate (**a**,**c**) and the signal amplitude estimate (**b**,**d**) on the SNR value for various parameter values *N* = 1024 and *N* = 2048, *f*_s_ = 3 GHz.

**Figure 6 sensors-22-03018-f006:**
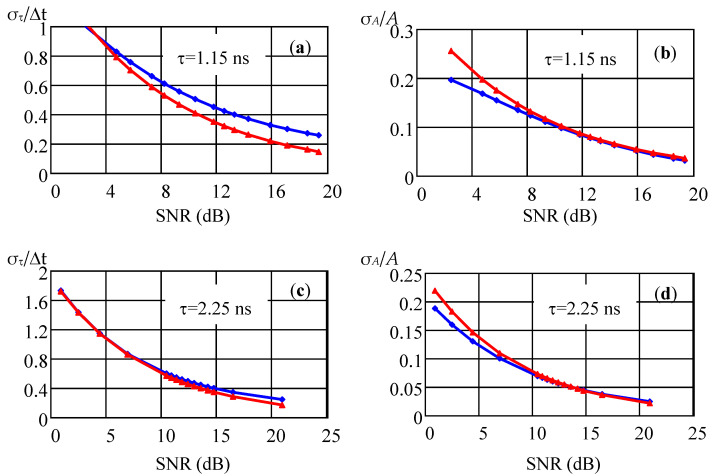
Dependences of the normalized standard deviations of the delay time estimate (**a**,**c**) and the signal amplitude estimate (**b**,**d**) on the SNR value for different values of the parameter τ = 1.15 ns and τ = 2.25 ns, *f*_s_ = 6 GHz.

**Figure 7 sensors-22-03018-f007:**
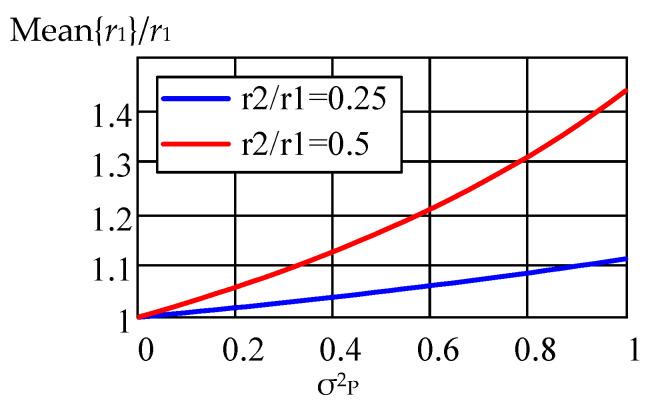
Dependence of the normalized value of the average value of the estimate *r*_1_ on the parameter σ^2^_P_ for two values of the ratio of radii *r*_2_/*r*_1_.

**Figure 8 sensors-22-03018-f008:**
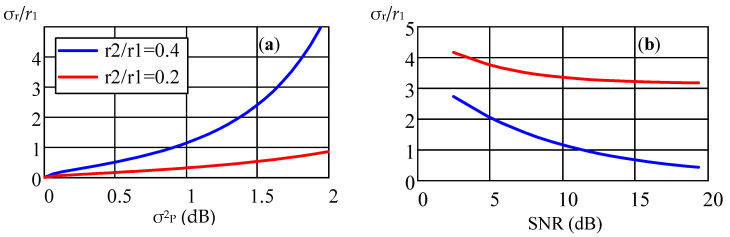
Dependences of the normalized standard deviation σ_r_/*r*_1_ of the estimate *r*_1_ on the parameter σ^2^_P_ (**a**) and the SNR ratio (**b**).

**Figure 9 sensors-22-03018-f009:**
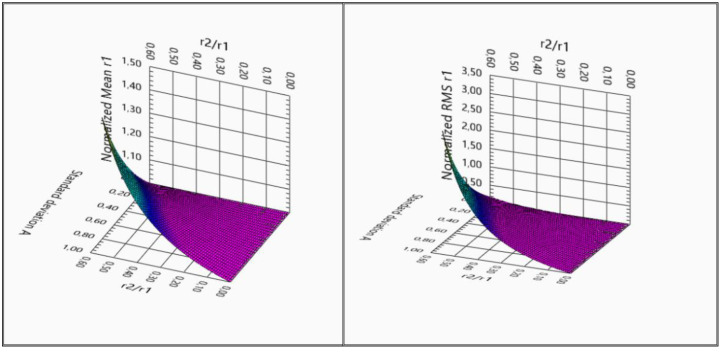
The 3D dependences of the normalized value of the mean value (**a**) and the normalized standard deviation of the estimate *r*_1_ (**b**) on the parameter σ^2^_P_ and the ratio of radii *r*_2_/*r*_1_.

## Data Availability

Not applicable.

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
