# Peer review of "RSS/TDoA-Based Source Localization in Microwave UWB Sensors Networks Using Two Anchor Nodesâ€"

_sensors, 2022, doi:10.3390/s22083018_

Round 1
Reviewer 1 Report
In this paper, the authors present an algorithm for optimal estimation of the amplitude and propagation delay time of an ultra-wideband radio signal in systems for passive location of fixed targets based on the hybrid RSS/TDoA method in two-dimensional space with two base stations. In general, the method in this paper is well described. However, this paper lacks of some important discussions, and the conclusions cannot be well supported by their simulations. Therefore, some revisions are required before acceptance.
- In Eq.(5) of section 2, the authors suppose that the noise is subject to Gaussian noise. In practice, there are many papers discussing the non-Gaussian noise, such as Class A noise [1], Alpha noise [2]. The authors should discuss the reason why to choose the Gaussian noise in Eq. (6). Besides, the authors should further discuss their method based on non-Gaussian noise. This can guide the readers’ research. Furthermore, this can make the authors’ work more interesting.
[1] X Zhang,et al.Parameter estimation of underwater impulsive noise with the Class B model.IET Radar, Sonar & Navigation,2020,Doi: 10.1049/iet-rsn.2019.0477.
[2] G Zbigniew,et al.Some Remarks on Maximum Likelihood Estimation in Alpha-Stable Environment.2020 Baltic URSI Symposium (URSI),2020,Doi: 10.23919/URSI48707.2020.9254052.
- Nowadays, the compressive sensing is a hot topic in the DoA, such as work in [3]. The reviewer wanders to know the potential of authors’ method with compressive sensing. Maybe, the authors can further discuss this issue in their conclusion.
[3] A Abdelreheem, EM Mohamed, H Esmaiel. Adaptive location-based millimetre wave beamforming using compressive sensing based channel estimation. IET Communications 13 (9), 1287-1296.
- In the section of simulation, the authors conduct the simulations. However, the simulations are too simple. That is to say, the conclusions cannot be validated based on authors’ simple experiments. The authors should further enhance their simulations. The comparison between authors’ method and traditional method should be carried out. Besides, the estimator performance is highly determined by SNR, which is neglected by authors in this paper. At this point, the authors should discuss this issue in detail in their simulations.
Author Response
Dear reviewer, please find in attached file our answers to your remarks.
Kind regards,
S.I., V.K.,V.B and A.F.

Reviewer 2 Report
The author provides a localization method combining RSS and TDOA, which further breaks through the limitations of these two traditional methods. There are several obvious issues that could be further considered if carefully revised.
1. The introduction part is lengthy, although it covers a lot of literature, but this is not a Review, it needs to give a logical and clear point of view.
2. In the introduction of the method, Bayesian estimation requires multiple results for continuous optimization, whether this will increase the time of UWB, and is not suitable for the application of single-shot signals?
3. The simulation process and results are not fully displayed. This part is very lacking, which directly leads to the difficulty of accepting the paper this time. The discussion of simulation or experiment needs to be further increased to verify the effectiveness of the method.
Author Response
Dear reviewer, please find in attached file our answers to your remarks.
Kind regards,
S.I., V.K., V.B and A.F.

Reviewer 3 Report
- The presented paper contains a very detailed theoretical analysis of the problem and its computer simulation. It seems to me that it would be useful to supplement the article with data from practical measurements and comparison of the obtained results with simulated data. Could the authors add this? If not, please comment on the reasons.
- I found some formal errors in the article, e.g. on line 484: Kramer - correctly Cramer, on line 438: R1 и R2 - correctly R1 and R2, etc. Please correct these errors.
Author Response

(The authors gave the same response as above.)

Round 2
Reviewer 1 Report
After revision, the paper is improved, and the authors has no more comments.
Author Response
Dear reviewer!
Thanks a lot for your kind help in our manuscript improvement.
Reviewer 2 Report
The author do well revision for the manuscript.
Author Response

(The authors gave the same response as above.)
